# Microstructural Features and Ductile-Brittle Transition Behavior in Hot-Rolled Lean Duplex Stainless Steels

**O. Takahashi [1], Y. Shibui [2,†], P.G. Xu [3] 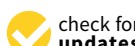, S. Harjo [3], T. Suzuki [4] and Y. Tomota [4,*,‡]**

[1]  NIDAK Co., 3641 Tezuna B-kogyoudanchi, Kamitezuna Asayama, Takahagi, Ibaraki 318-0004, Japan; ndktaka@atlas.plala.or.jp

[2]  Graduate Student of Ibaraki University, 4-12-1 Nakanarusawa, Hitachi, Ibaraki 316-8511, Japan; yohei.shibui@hayashiseiki.co.jp

[3]  Japan Atomic Energy Agency, 2-4 Shirakata, Tokai, Naka, Ibaraki 319-1195, Japan; xu.pingguang@jaea.go.jp (P.G.X.); stefanus.harjo@j-parc.jp (S.H.)

[4]  Graduate School of Science and Engineering, Ibaraki University, 4-12-1 Nakanarusawa, Hitachi, Ibaraki 316-8511, Japan; tetsuya.suzuki.corong@vc.ibaraki.ac.jp

\*  Correspondence: TOMOTA.Yo@nims.go.jp or yo.tomota.22@vc.ibaraki.ac.jp

†  Now at Hayashi Seiki Seizo Co., Ltd.

‡  Now at National Institute of Advanced Industrial Science and Technology.

**Abstract:** The characteristics of texture and microstructure of lean duplex stainless steels with low Ni content produced through hot rolling followed by annealing were investigated locally with electron backscatter diffraction and globally with neutron diffraction. Then, the ductile–brittle transition (DBT) behavior was studied by Charpy impact test. It is found that the DBT temperature (DBTT) is strongly affected by the direction of crack propagation, depending on crystallographic texture and microstructural morphology; the DBTT becomes extremely low in the case of fracture accompanying delamination. A high Ni duplex stainless steel examined for comparison, shows a lower DBTT compared with the lean steel in the same crack propagating direction. The obtained results were also discussed through comparing with those of cast duplex stainless steels reported previously (Takahashi et al., *Tetsu-to-Hagané,* 100(2014), 1150).

**Keywords:** duplex stainless steel; ductile-brittle transition; neutron diffraction; texture; microstructure; electron backscatter diffraction

---

## 1. Introduction

Duplex stainless steels consisting of ferrite ($\alpha$) and austenite ($\gamma$) have been widely used particularly in corrosive environments [1]. The tensile strength is mainly dependent on the volume fraction of $\gamma$ [2–5], which is usually adjusted to be approximately 50% by controlling the annealing temperature. In contrast, the impact toughness is influenced by several factors and often becomes problematic [6–9]. Similarly to ferritic steels, the duplex stainless steels exhibit ductile to brittle transition (DBT) behavior, despite $\gamma$ phase fractures in a ductile manner even at a low temperature. The DBT temperature (DBTT) determined by Charpy impact test has been employed for primary evaluation of toughness.

In general, large-scaled mechanical products are produced by casting whereas medium or small ones by plastic forming. As has been reported by the present authors for cast duplex stainless steels [10–12], the DBTT obtained for specimens made from a keel block test sample is often different from that obtained for specimens prepared from a real large-scaled product itself, meaning that it is difficult exactly to evaluate the toughness for integrity insurance by a keel block casting method.

As is summarized in Figure 1, the difference in DBTT between a real product made by centrifugal casting and a keel block is approximately 20 °C; if the impact energy of 130 J was required at the room temperature (RT), the result for the keel block test sample would fail although that obtained by the real product would satisfy.

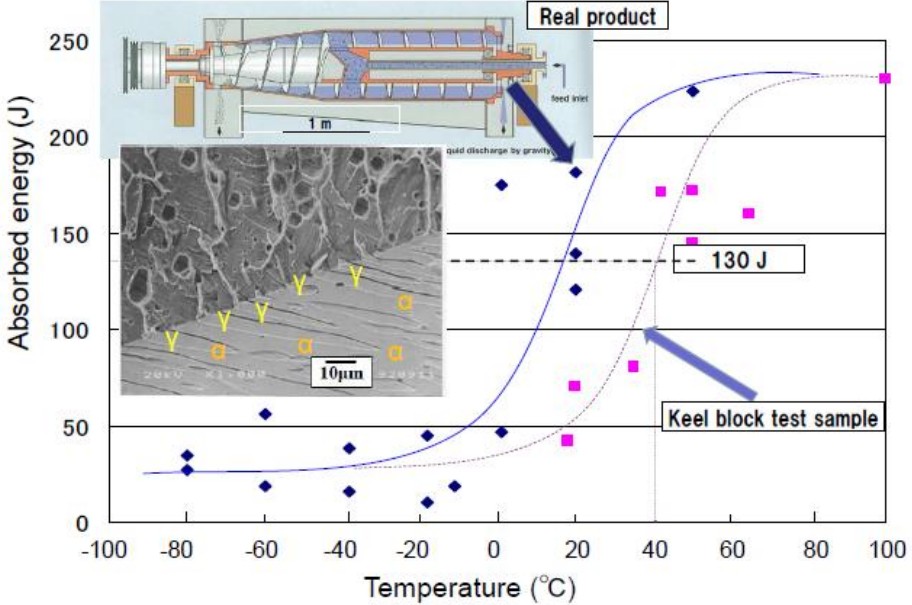

**Figure 1.** Comparison of ductile–brittle transition (DBT) curves obtained for a cast duplex stainless steel product (inserted) with that for keen-block test sample after References 10–12. A two-plane observation for fracture surface and microstructure [11] was also inserted to show the cleavage fracture of ferrite grains accompanied with ductile fracture of austenite grains at −196 °C.

In Figure 1, another inset shows a result of two-plane observation via scanning electron microscopy (SEM): one plane shows α-γ microstructure and the other fractography revealing {100} quasi-cleavage of α grains and ductile rupture of γ grains at −196 °C where {100} fracture plane was confirmed by an etch-pit method [10,11]. The DBTT was believed to be affected by the morphology and size of fracture facet (grain), texture, phase volume fraction and harmful impurities [10–12]. Recently, the so called "lean stainless steel with low Ni content" has been more frequently used, and their DBTT has become higher with decrease in Ni content.

In this study, the DBTT and fracture behavior of a lean duplex stainless steel as hot-rolled and annealed condition was studied, and the influence of crack propagation direction with respect to the anisotropic microstructure and texture was discussed. In addition, the DBT behavior of a hot-rolled high Ni bearing duplex stainless steel was examined in order to deepen the general understanding on DBT behavior of the duplex stainless steel. The obtained results were also discussed referring to the corresponding results for cast steels reported previously [11].

## 2. Experimental Procedures

Duplex stainless steels used in this study were so-called lean duplex stainless steel, LDX2101 produced by Outokumpu Co., Avesta, Sweden (hear after called LNi steel) and SUS329J4L of Japanese Industrial Standard designation (JIS) made by Nippon Yakin Kogyo Co. Ltd., Kawasaki, Japan (HNi steel) for comparison. The chemical compositions of these two steels (LNi and HNi) are given in Table 1. These steels were supplied as hot-rolled plates. Then, LNi steel plates were annealed at 1050 °C for 6 min followed by water-quenching (hereafter called sample LNi or just LNi), whereas some of the plates were annealed at 1100 °C for 6 h followed by water-quenching in order to obtain coarse microstructure (LNi-A). On the other hand, HNi steel plates were annealed at 1060 °C

for 6 min followed by water quenching (HNi) and some plates were annealed at 1100 °C for 6 h also followed by water-quenching (HNi-A). Consequently, four types of samples (LNi, LNi-A, HNi and HNi-A) were prepared for investigation.

**Table 1.** Chemical compositions of duplex stainless steels used (mass%).

| Steel | C | Si | Mn | P | S | Cr | Ni | Mo | Cu | N |
|-------|------|------|------|-------|--------|------|------|------|------|------|
| LNi | 0.023 | 0.71 | 4.92 | 0.022 | 0.001 | 21.4 | 1.57 | 0.41 | 0.22 | 0.23 |
| HNi | 0.019 | 0.41 | 0.78 | 0.027 | <0.001 | 24.9 | 6.59 | 3.18 | - | 0.17 |

Scanning electron microscopy (SEM) microstructures of these samples were observed with a field emission-scanning electron microscope, HITACHI S-4300SE (Hitachinaka, Japan) operated at 20 kV for samples finished by electrolytic polishing at Ibaraki University. Electron backscatter diffraction (EBSD) analyses were performed using TSL OIM analysis software (Ver. 5.0).

To obtain the bulk-averaged orientation distribution function (ODF), time-of-flight (TOF) neutron diffraction measurements were carried out for 10 mm cubic samples at an engineering neutron diffractometer, BL 19 (TAKUMI) of the Materials and Life Science Experimental Facility (MLF) at the Japan Proton Accelerator Research Complex (J-PARC) [13,14]. The sample was two-axes ($\chi$, $\phi$) rotated 35 times to collect totally 525 neutron diffraction profiles with different scattering vectors using an Euler cradle [15,16]. The obtained TOF neutron profiles (with a lattice plane spacing (*d*) range of 0.05~0.22 nm, including 18 diffraction peaks of $\gamma$ and 15 of $\alpha$) were analyzed using MAUD (Materials Analysis Using Diffraction) software [17] to reconstruct the multiphase pole figures and simultaneously to determine the globally averaged $\gamma$ volume fraction [18], which has been recommended for a textured steel [15,16]. Such pole figures were further employed to quantitatively calculate multiphase ODFs.

JIS V-notch Charpy impact test specimens were machined along two direction as illustrated in Figure 2, in which X, Y and Z axes were defined parallel to the rolling direction (RD), the transverse direction (TD) and the normal direction (ND) of a plate, respectively. In total, four kinds of specimens, XZ, XY, YZ and YX were prepared. The first capital refers to the macroscopic fracture plane normal and the second to the crack propagation direction related to the notch; for example, XZ stands for the case that a crack propagates on the X plane along the Z direction. A Charpy impact tester made by Tokyo Koki Co, (Tokyo, Japan) was used for the impact tests at NIDAK Co. (Takahagi, Japan) where test temperature was controlled using ethyl alcohol cooled with liquid nitrogen for lower temperatures than RT and oil heated for higher temperatures.

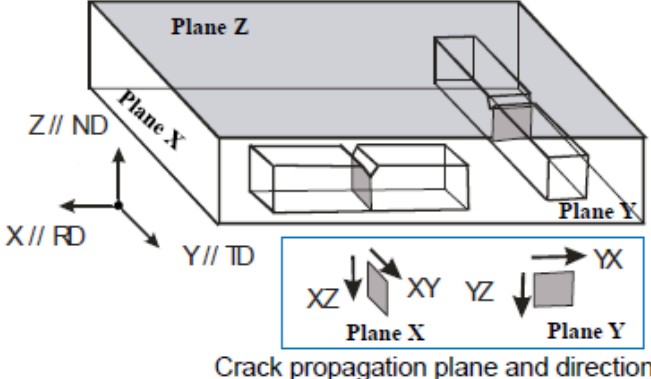

**Figure 2.** V-notch Charpy impact test specimen preparation from hot-rolled and annealed plates, where the definition of plane and direction was illustrated.

## 3. Results and Discussion

### 3.1. Microstructural Features

SEM microstructures observed on the planes X, Y and Z for LNi are presented in Figure 3. These microstructures consist of $\alpha$ and $\gamma$ phases, being similar to the cases of the previously reported cast steels although the morphology and grain size of the $\alpha$ phase are significantly different from each other; both of the $\alpha$ and $\gamma$ grains are elongated along the rolling direction (*X*-axis) with similar grain sizes, which are commonly observed for LNi and HNi.

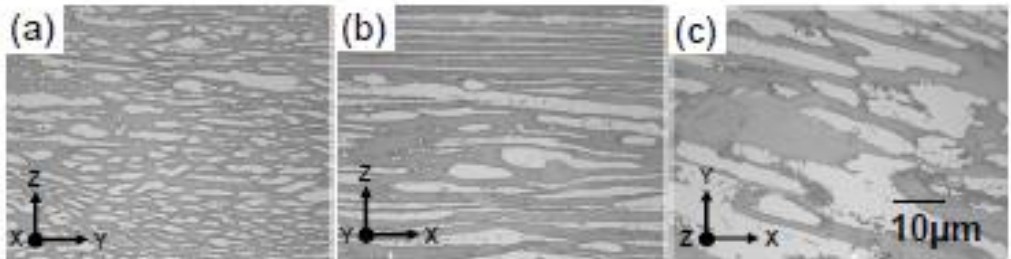

**Figure 3.** SEM micrographs of LNi steel: (**a**) observation on plane X, (**b**) Y and (**c**) Z. Concerning the definition of plane, see Figure 2.

The {100} cleavage fracture at a cryogenic temperature is influenced by so-called "effective grain size" corresponding to fracture facet firstly proposed by Matsuda et al. [19–21] for tempered martensite steels. This concept has been generally applied to body centered cubic (bcc) metals [22]. Though the effective grain size for a duplex stainless steel might be difficult to be determined [11,12], the $\alpha$ grain size must be an important factor for DBT behavior. Then, the phase and IPF maps for the $\alpha$ phase observed on the X plane of LNi and LNi-A are presented in Figure 4. As can be seen, the grains were a little bit spheroidized for LNi-A due to the occurrence of grain coarsening, though the volume fraction and texture were hardly changed by annealing conditions. Here, the shape of $\alpha$ grain is assumed ellipsoidal, and the mean lengths along the X, Y and Z axes were respectively measured by a line intercepting method for SEM micrographs. The results obtained for LNi were 30.6, 10.3 and 1.65 μm for X, Y and Z direction, respectively. The grain shape was slightly changed in LNi-A; the mean grain lengths measured were 18.1, 10.0 and 2.86 μm for X, Y and Z direction, respectively. The microstructures of HNi and HNi-A were similar with those of LNi and LNi-A, respectively. Considering that it must be important to check the crystal orientations of $\alpha$ grains in three orthogonal directions, the IPF maps obtained on the X, Y and Z planes for the crystal orientation in the X, Y and Z directions were examined. An example of such observation is given in Figure 5 for LNi. It should be noted that preferential <100> texture along the *Z*-axis is remarkably developed.

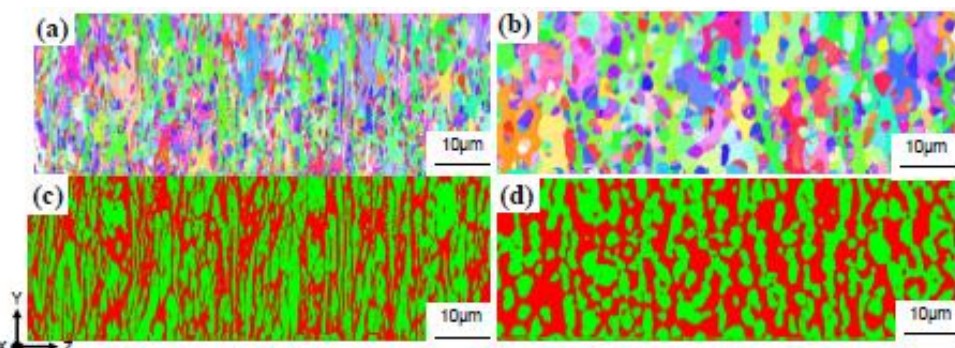

**Figure 4.** Inverse pole figure (IPF) maps (**a**,**b**) and phase maps (**c**,**d**) for plane X of steels LNi (**a**,**c**) and LNi-A (**b**,**d**), where red and green areas are corresponding to ferrite ($\alpha$) and austenite ($\gamma$), respectively. The standard color triangle IPF for crystal orientation is presented in Figure 4.

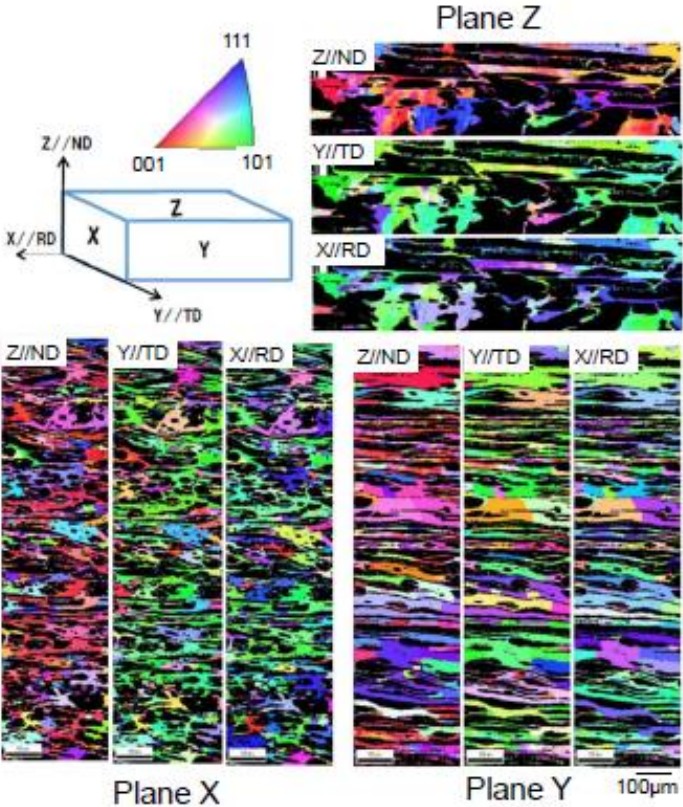

**Figure 5.** Inverse pole figure (IPF) maps related to three directions for the ferrite ($\alpha$) phase in steel LNi observed on three orthogonal planes X, Y and Z. The insets are the definition of plane and direction and the standard color triangle IPF for crystal orientation, respectively.

### 3.2. Characteristics of Texture Determined by Neutron Diffraction

To obtain more quantitative and global information about texture and true phase volume fractions, neutron diffraction profiles from 525 directions (scattering vectors) were collected and analyzed using MAUD software [17], and an example of Rietveld combined analysis for texture and phase volume fraction is given for LNi in Figure 6. The volume fraction of the $\gamma$ phase obtained from the combined analysis was 51.34 ($\pm$0.05)%, which was sufficiently precise after the quantitative evaluation of the global phase textures. The pole figures were re-constructed from these diffraction analysis data and presented in Figure 7. The Kurdjumov–Sachs (K-S) orientation relationship between the $\alpha$ and $\gamma$ phases [23] has been confirmed through the microstructure observation of duplex stainless steels using transmission electron microscopy, so far [24–26], and such an orientation relationship was suggested partially for LNi from Figure 7 (see arrows). It is believed that the present duplex stainless steels solidified firstly as $\alpha$ phase and then that $\gamma$ grains precipitated having the K-S relationship with the $\alpha$ matrix upon cooling. During hot rolling, work hardening, dynamic recovery and recrystallization in $\alpha$ and $\gamma$ grains and probably additional $\gamma$ precipitation took place. Upon cooling after the hot rolling, the static recovery and recrystallization must occur. Then, the static recovery, recrystallization, grain growth and coalescence would happen during annealing. Interestingly, even after such a complicated heat history, the K-S relationship was partially kept between the $\alpha$ and $\gamma$ phases in Figure 7. As shown in Figure 4, the grain growth and coalescence seem to take place in LNi-A. Figure 8 shows the result of MAUD fitting for LNi-A; the volume fraction of $\gamma$ was 45.90 ($\pm$0.05)%. The pole figures in Figure 9 suggest that the texture of $\gamma$ became weaker compared with that for LNi because of annealing at a higher temperature for a longer time. In order to obtain more detailed texture information, the ODF figures for LNi and LNi-A are depicted in Figure 10. As seen, the texture is relatively weaker for LNi-A than LNi caused by grain coarsening. Here, it should be noted that {100}<110> component marked by

arrows in Figure 10 is very intense for the α phase. This implies that {100} cleavage fracture takes place easily along the Z-plane.

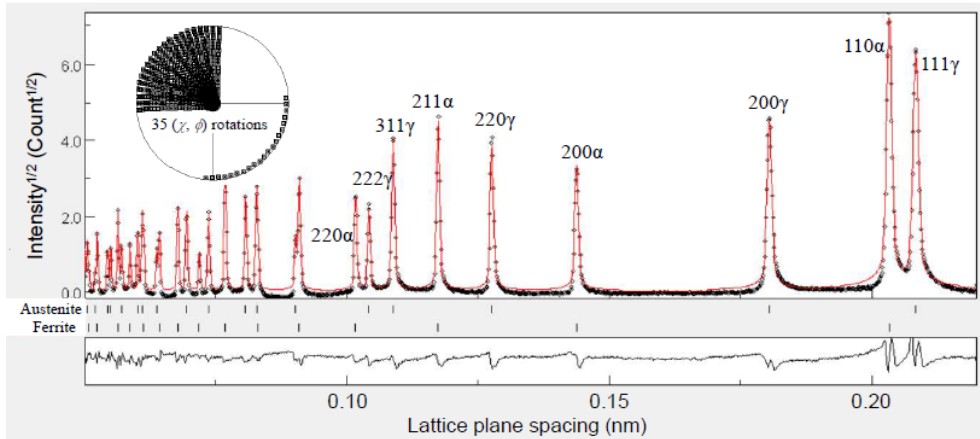

**Figure 6.** Rietveld analysis result of texture and phase volume fraction for steel LNi, using TOF neutron diffraction profiles obtained from 525 directions (scattering vectors) shown in the inset.

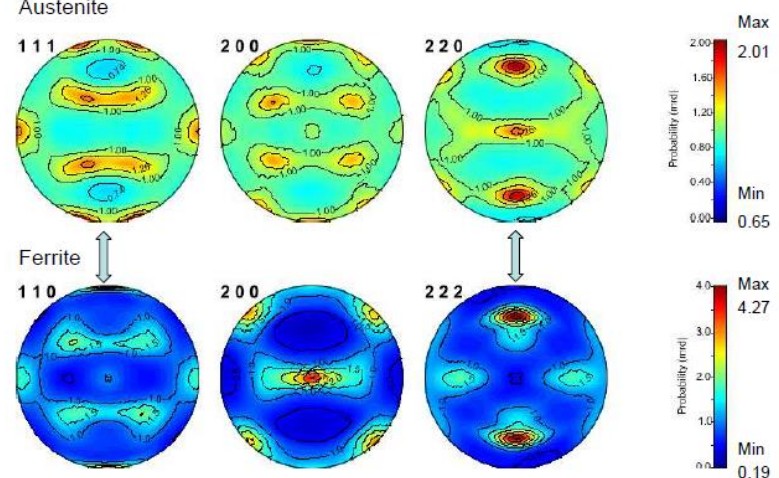

**Figure 7.** Pole figures of austenite (γ) and ferrite (α) obtained by neutron diffraction for LNi.

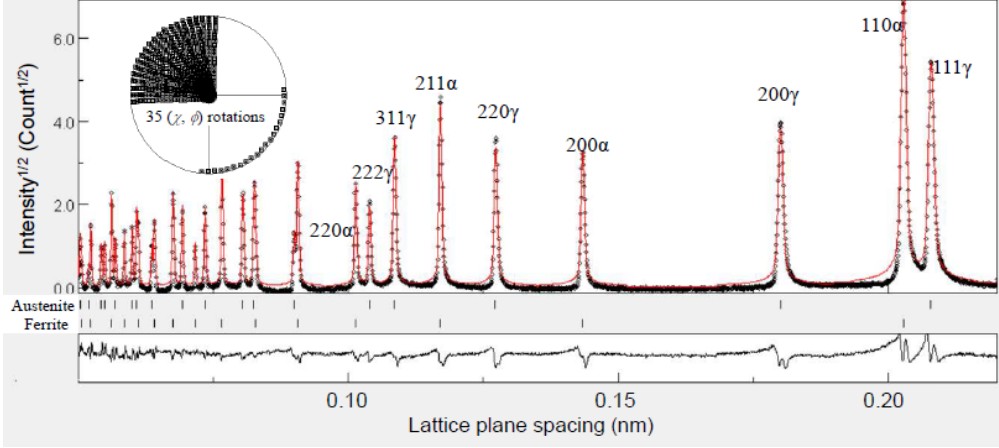

**Figure 8.** Rietveld analysis result of texture and phase volume fraction for steel LNi-A, using TOF neutron diffraction profiles obtained from 525 directions (scattering vectors) shown in the inset.

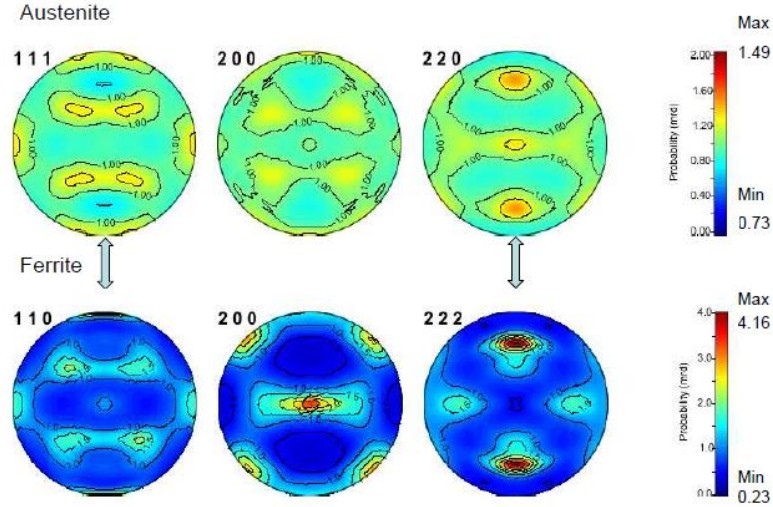

**Figure 9.** Pole figures of austenite and ferrite obtained by neutron diffraction for steel LNi-A.

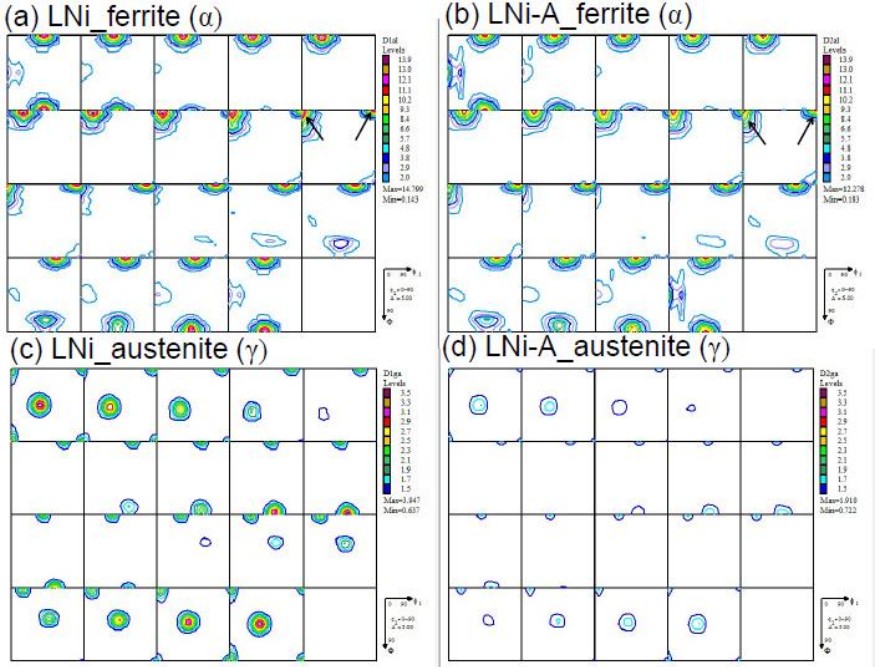

**Figure 10.** DFs for ferrite ($\alpha$) (**a**,**b**) and austenite ($\gamma$) (**c**,**d**) calculated using the reconstructed pole figures obtained from 525 neuron diffraction profiles for LNi (**a**,**c**) and LNi-A (**b**,**d**).

### 3.3. Influence of Crack Propagation Direction on DBT Behavior

The influence of crack propagation direction on the DBT curve obtained by Charpy impact tests for LNi is presented in Figure 11; crack propagation plane is perpendicular to the X axis (X-plane) in (a) whereas to the Y axis (Y-plane) in (b). As can be seen, the DBT behavior depends apparently on the crack growth (or notch) direction; the DBTT for specimen XZ of LNi is significantly lower than specimen XY in Figure 11a. Similar trend is observed for LNi-A with coarse grains but the difference between specimens XZ and XY is small. A higher DBTT for specimen XZ of LNi-A compared with that for XZ of LNi is postulated to be caused by coarser grains with higher connectivity (see Figure 4), and weaker texture of the $\alpha$ phase. Similar trend is found in the case of crack propagation along the Y plane in Figure 11b. Here, DBTT was determined by estimating the temperature to show the mean impact energy between the upper and the lower shelves. The upper shelf energy

is found to be lower in Figure 11b than in Figure 11a, particularly for the specimen YZ and YX of LNi-A. The lower ductility at temperatures higher than DBTT must be attributed to the anisotropic microstructure. Consequently, when the crack propagation direction is parallel to the Z direction, the DBTT is commonly lower both in (a) and (b), suggesting a characteristic fracture mechanism along the Z-axis; the occurrence of delamination will be explained later.

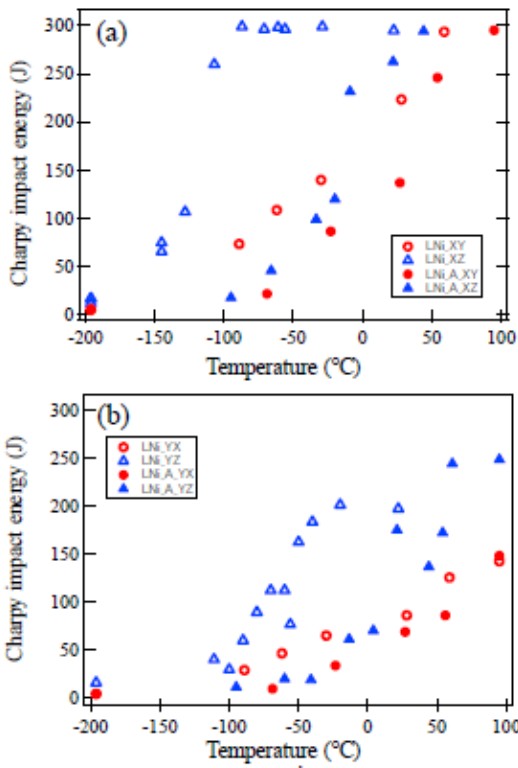

**Figure 11.** DBT curves for the XY and XZ specimens (**a**) and those for YX and YZ (**b**) of LNi and LNi-A.

The DBT curves for HNi were also measured for comparison, and the results obtained are presented in Figure 12. The overall trend observed in this figure is nearly the same with that in Figure 11; the DBTT was lower when the crack propagated parallel to the Z direction. It is found that Ni alloying lowers DBTT and increases the upper shelf energy. The influence of crack propagation direction is relatively smaller for HNi than LNi. It should be noted that the DBTT for specimen XZ of HNi was lower than −196 °C indicating extremely high toughness.

The comparison between LNi and HNi is summarized in Figure 13, in which the results for cast duplex stainless steels reported previously [10–12] are also presented. Consequently, the DBTT of duplex stainless steel depends greatly on the crack propagation direction for the hot-rolled plates. The reason why DBTT is so sensitive to the crack propagation direction is speculated to be related to the texture and microstructural features. The difference between the hot-rolled plates and cast steels must be mainly due to the α grain size and connectivity; as reported in Reference [11], the α grains observed separately in a two-dimensional observation were frequently connected three-dimensionally, which was revealed by a serial sectioning observation method. The DBT behavior in the cast steels was influenced by casting conditions such as metal- or sand-mold, casting temperature, etc., but this influence is much small compared with the influence of crack propagation direction in the forged duplex stainless steels. The increase of Ni content is also observed to lower DBTT in both of cast and forged duplex stainless steels, similarly to the general trend observed in ferritic steels [27,28]. The effect of Ni on DBTT is not clear and still open for discussion.

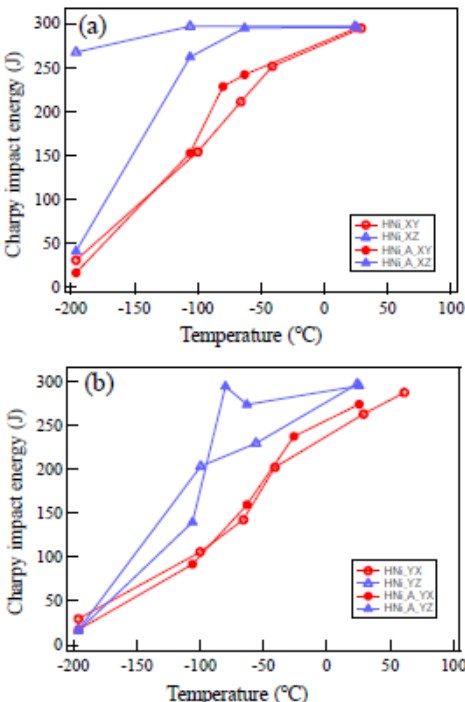

**Figure 12.** DBT curves for XY and XZ specimens (**a**) and those for YX and YZ (**b**) of HNi and HNi-A.

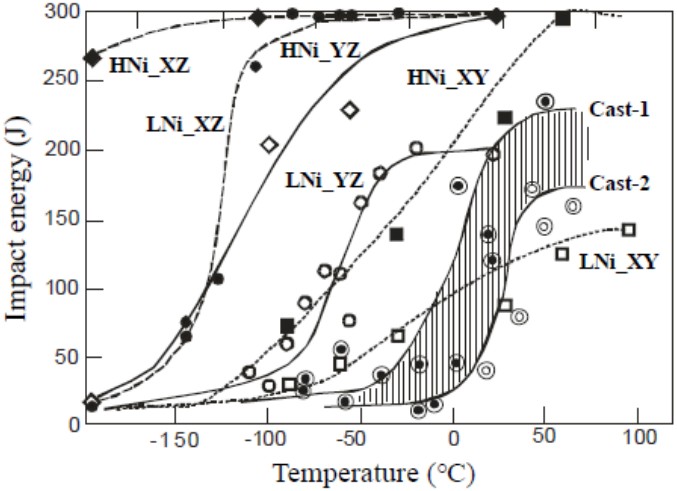

**Figure 13.** Comparison of DBT behavior in duplex stainless steels with different compositions and microstructures.

### 3.4. Effect of Microstructure and Texture on Low Temperature Fracture Behavior

Typical outlooks of fractured specimens are presented in Figure 14. As can be found in (a) and (b), sub-cracks perpendicular to the main crack growth direction were observed for specimen XZ of LNi fractured at −196 °C. That is, so-called "delamination" occurred for specimen XZ of LNi, leading to lower DBTT. Contrary to this case, fracture surface is almost flat for specimen YX of LNi in (c) and (d), although short sub-cracks are observed. The sub-cracks were found always along the Z plane and classified into two types, S1 and S2. Fractographic features observed at a higher magnification for these two types are presented in Figure 15a,b, respectively; type S1: the sub-crack plane normal is perpendicular to the main crack propagation direction and type S2: sub-crack plane normal is parallel to the main crack propagation direction (see illustrations in Figure 15). As would be expected,

the type S2 is much more effective to hinder the main crack propagation which is called "delamination toughening" [29–31]. That is, a specimen with a notch oriented towards the crack propagation along the Z-axis shows lower DBTT due to the occurrence of delamination. Fracture surface with delamination observed at a higher magnification is presented in Figure 16, which consists of cleavage fracture of α grains and ductile fracture of γ grains. Examples of crystallographic characteristics of a sub-crack examined with SEM/EBSD on the sectioned planes of fractured specimens are presented in Figure 17; (a) shows several sub-cracks in the vicinity of a main crack. It is also observed that fracture surface perpendicular to the main crack (see arrows in the figure). Sub-cracks were examined with EBSD in detail and a typical example is shown in Figure 17b–d, revealing that the sub-crack took place along α{100} plane perpendicular to the Z-axis (see b). Here, the sub-crack is speculated to stop its propagation at the α/γ interface. Consequently, the {100} cleavage on the Z plane occurs frequently even in the case that a main crack plane is perpendicular against it. Hence, although specimens ZX or ZY were not examined because it was difficult to prepare such a specimen (see Figure 1), the DBTT for such specimen is postulated extremely high due to the occurrence of preferential {100} cleavage fracture along the Z plane without accompanying any sub-cracks.

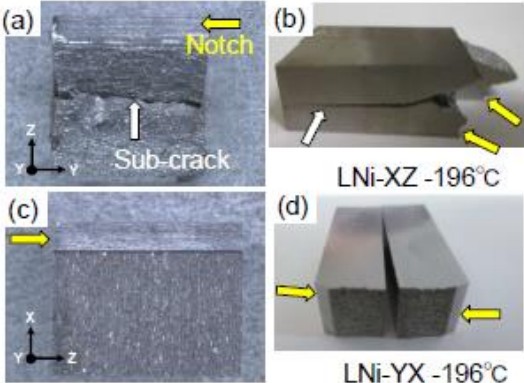

**Figure 14.** Characteristics of fractured specimens of LNi: XZ (**a**,**b**) and YX (**c**,**d**): (**a**,**c**) are fracture surface whereas (**d**,**e**) two separated parts of fractured specimen. Concerning the microscale for these photographs, refer to the dimensions of JIS V-notch Charpy impact test specimen (10 × 10 × 55 mm).

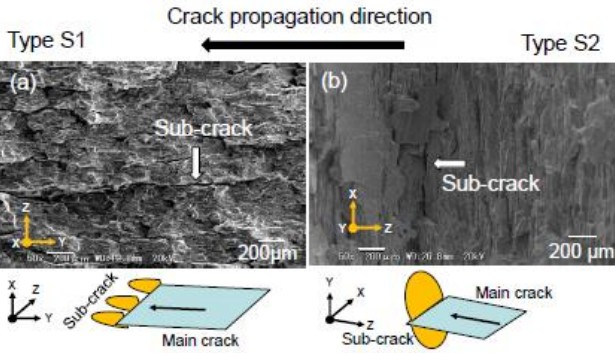

**Figure 15.** Fractography of specimens of LNi fractured at −196 °C: (**a**) XY (type S1) and (**b**) YZ (type S2).

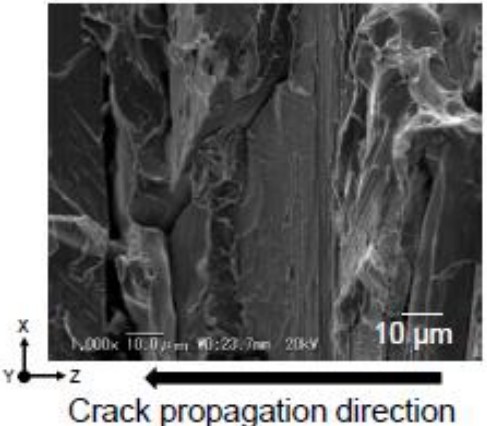

**Figure 16.** Features in fractography of specimen YZ of LNi fractured at −95 °C

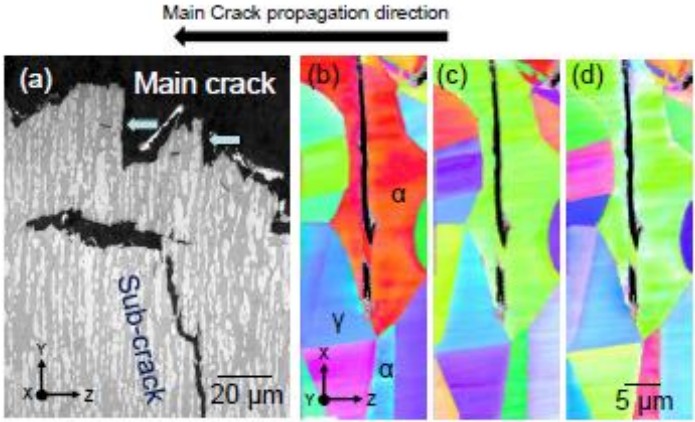

**Figure 17.** Features of sub-crack observed: (**a**) specimen XZ of LNi-A fractured at −196 °C and (**b**–**d**) IPF maps for specimen YZ of LNi fractured at −95 °C, where (**b**) Z(//ND), (**c**) Y(//TD) and (**d**) X(//RD). Concerning the crystal orientation color for IPF map, see Figure 5.

### 3.5. Role of Ductile γ Phase and Effect of α Grain Size on DBTT

The DBTT determined from Figures 11 and 12 are summarized in Table 2. A clear trend is found that the DBTT becomes higher for specimen XZ, YZ, XY and YX in order, except for the cases of YZ and XY of LNi-A. When the main crack propagates along the Z direction either on the X or Y plane, type S2 sub-cracks occurs leading to lower DBT. The difference between the X and Y planes is believed to stem from the difference in the mean free length of α grain; the facet width with {100} cleavage fracture in an α grain is smaller for the X plane (i.e., the length along *Y*-axis: 10.3 μm for LNi as described in Section 3.1) than that for the Y plane (the length along *X*-axis 30.6 μm). The effective gran size for fracture unit employed for ferritic steels [16–19,32], is difficult to apply to the present steels. For the cast duplex stainless steels [10–12], small γ grains were dispersed within a huge α grain and hence the DBTT was summarized assuming the α grain size as the effective grain size (ignoring the presence of γ grains) in Figure 18. Since the ductile fracture of γ grains hinders the propagation of cleavage fracture of α grains as presented in Figure 1, the DBTT is lower in comparison with the trend for forged ferritic steels [19,33], which has been extended to a sub-micron meters range for ultra-fine-grained steels [32]. In the case of the present hot-rolled duplex stainless steels, the grains are extremely elongated. Hence, the mean value of two-orthogonal lengths was tentatively employed as the effective grain size and the results obtained for LNi were added in Figure 18. Concerning the other specimens of LNi-A, HNi and HNi-A, refer to Table 2 taking into similar effective grain sizes into consideration. Though significantly anisotropic features for fracture at cryogenic temperatures exist, it would be confirmed that the γ grains work to lower DBTT, similarly to the cases of martensitic Ni

steels for cryogenic use, in which the volume fraction, stability and grain shape of the γ grains have been known to affect DBTT [34,35]. The ductile fracture of γ grains is, therefore, believed to lower DBTT for duplex stainless steel, but it is not so crucial compared with the influence of texture, size and connectivity of α grains.

**Table 2.** Influence of crack propagation (notch) plane and direction on DBTT (°C) for LNi and HNi steels.

| Steel\Notch Plane and Direction | XZ | YZ | XY | YX |
|---|---|---|---|---|
| LNi | −130 | −65 | −25 | −5 |
| LNi-A | −20 | +30 | +25 | +35 |
| HNi | ≤196 | −105 | −95 | −55 |
| HNi-A | −150 | −105 | −115 | −65 |

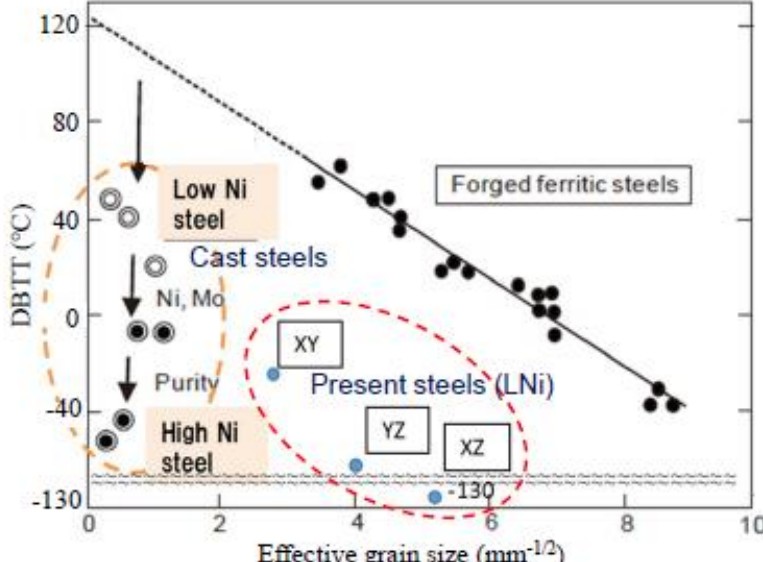

**Figure 18.** DBTT as a function of the effective grain size (the data for forged ferritic steels were replotted from References [22,33]).

## 4. Conclusions

The texture, microstructural characteristics, and ductile–brittle transition (DBT) behavior in lean duplex stainless steels bearing low Ni content produced through hot rolling followed by annealing were studied. Additionally, a high Ni duplex stainless steel was also examined for discussing the influence of Ni content on DBT. The main conclusions obtained in this study would be summarized as follows:

(1) The textures of the lean duplex stainless steels consisting of ferrite (α) and austenite (γ) were characterized locally by using EBSD and quantitatively evaluated by the TOF neutron diffraction, the results of which were analyzed using MAUD software. As results, intense {100}<110> component in α was confirmed and the γ volume fraction was precisely determined with satisfactory texture-correction.

(2) The connectivity of elongated grains and the preferred <100> orientation in the α phase along the normal direction of a steel plate play a critical role on the anisotropic DBT behavior of these duplex stainless steels.

(3)    The DBT temperature (DBTT) obtained by V-notch Charpy impact tests is strongly affected by the direction of crack propagation; the cases of crack propagation in the normal direction shows lower DBTT accompanying sub-cracks, so called "delamination toughening".

(4)    In the same crack propagating direction, a higher Ni bearing steel shows a lower DBTT compared with a lower Ni bearing, i.e., lean steel.

(5)    The dispersed $\gamma$ grains hinder the propagation of $\alpha$ cleavage fracture, leading to lower DBTT.

**Author Contributions:** Data curation, O.T., Y.S., P.G.X. and S.H.; Investigation, O.T., Y.S. and Y.T.; Methodology, O.T., Y.S., P.G.X. and S.H.; Project administration, O.T., T.S. and Y.T.; Supervision, T.S. and Y.T.; All authors have read and agreed to the published version of the manuscript.

**Funding:** This research received no external funding.

**Acknowledgments:** Experimental assistance by M. Matsushima and M. Yabe (NIDAK Co.), H. Sato (technical staff of Ibaraki University) and K. Shinba (student of Ibaraki University; now at Ono Medical Co.) are highly acknowledged. Neutron diffraction measurements were performed through a JAEA project program 2014P0102 at J-PARC MLF.

**Conflicts of Interest:** The authors declare no conflict of interest.

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
