# Peer review of "Microstructural Features and Ductile-Brittle Transition Behavior in Hot-Rolled Lean Duplex Stainless Steels"

_qubs, doi:10.3390/qubs4010016_

Round 1
Reviewer 1 Report
The paper is well written and should be published. I only have a few minor suggestions that would improve clarity:
1) In the experimental procedures it is described that some samples were annealed 6 minutes followed by water-quenching, whereas other samples were annealed for 6 hours. No information is provided about how the later set of samples was cooled down. This should be rectified. Even if it is not part of the current analysis, the influence of the cooling speed on the properties of the sample might come into focus in the future.
2) The meaning of the arrows in figure 10 a) and b) is explained in the text only. It should also be included in the figure caption.
Author Response
qubs-736040
Response to reviewers’ commentaries
Accepting the reviewers’ comments, we revised the manuscript. The changes made are presented in green color throughout the revised text. The main revisions are explained below.
Reviewer 1
Comments and Suggestions for Authors
The paper is well written and should be published. I only have a few minor suggestions that would improve clarity:
- In the experimental procedures it is described that some samples were annealed 6 minutes followed by water-quenching, whereas other samples were annealed for 6 hours. No information is provided about how the later set of samples was cooled down. This should be rectified. Even if it is not part of the current analysis, the influence of the cooling speed on the properties of the sample might come into focus in the future.
Response: The other samples were commonly water-quenched after annealing. Then, the sentence, “followed by water-quenching”, was added in the new version.

2) The meaning of the arrows in figure 10 a) and b) is explained in the text only. It should also be included in the figure caption.
Response: We added the explanation in the figure caption.

Reviewer 2 Report
Generally this is an interesting and well presented piece of scientific work. The methods are explained effectively and the arguments presented are sound. I have no concerns over the content except for the following comments:
There is some odd use of English language within the paper, in-particular informal wording. I suggest having a native English speaker correct the text.
Avoid use of first person 'we'.
Were the neutron diffraction results analysed for residual stress?
Where was the EBSD performed?
Scale bar missing on Figure 14
Author Response
qubs-736040
Response to reviewers’ commentaries
Accepting the reviewers’ comments, we revised the manuscript. The changes made are presented in green color throughout the revised text. The main revisions are explained below.
Reviewer 2
Comments and Suggestions for Authors
Generally this is an interesting and well presented piece of scientific work. The methods are explained effectively and the arguments presented are sound. I have no concerns over the content except for the following comments:
There is some odd use of English language within the paper, in-particular informal wording. I suggest having a native English speaker correct the text.
Avoid use of first person 'we'.
Response: We apologize for that inconvenience. We checked the manuscript very carefully and improved language quality. The sentence beginning from “We” was changed at two places.
Commentary: Were the neutron diffraction results analysed for residual stress?
Response: Residual stresses were not evaluated by neutron diffraction because impact specimens were machined from large-scaled plates after heat treatment.
Commentary: Where was the EBSD performed?
Response: The EBSD measurements were performed at Ibaraki University, which was added in the text.
Commentary: Scale bar missing on Figure 14.
Response: The precise scale bar cannot be given because these were pictured using a close-up camera. Instead, we added the explanation in figure caption “The size of fracture surface was approximately 10 mm by 8 mm as is postulated from the dimensions of the impact specimen used.
Some minor modifications were also made as shown in blue color throughout the revised text.
